# Leveraged Asymmetric Loss with Disambiguation for Multi-label Recognition with One-Positive Annotations

## Abstract

In the problem of multi-label learning from single positive labels (SPL), we learn the potential multiple labels from one observable single positive annotation. Despite many efforts to solve this problem, an effective algorithm with sound theoretical understanding is still in need. In this paper, we propose a novel loss function for the SPL problem, called *leveraged asymmetric loss with disambiguation (LASD)*, where we introduce a pair of leverage parameters to address the severe negative-positive imbalance. From the theoretical perspective, we analyze the SPL problem, for the first time, from the perspective of risk consistency, which links the SPL loss with losses for ordinary multi-label classification. We prove the consistency of our proposed LASD loss to the cost-sensitive Hamming loss, which provides guidance to the empirical choice of our proposed leverage parameters. In experiments, we demonstrate the effectiveness of our proposed LASD loss function over other state-of-the-art methods and empirically verify our theoretical results.

## 1 Introduction

Different from standard multi-class classification, where each instance is tagged with one target label, multi-label classification (Liu et al., 2022; Li et al., 2022) allows an instance to have multiple labels and thus is applicable to wider real-world scenarios. For example, a picture can contain multiple objects (Lanchantin et al., 2021; Hu et al., 2021), a sentence can express multiple emotions (Huang et al., 2021; Fei et al., 2020), and a song can belong to multiple genres (Shrivastava et al., 2020; Pellegrini & Masquelier, 2021). Despite the wide applications of multi-label learning, the existence of multi-labels further increases the difficulty of annotating high-quality labels (Deng et al., 2014). On the one hand, label annotation can be extremely laborious and costly (Deng et al., 2014). On the other hand, the small objects or rare classes are often inevitably ignored by the human annotators (Liu et al., 2021; Wolfe et al., 2005). To deal with such problems, researchers loosen the requirements of label and propose the "Partial Multi-Label" (PML) paradigm, where the label of each instance can be the subset of its complete label set (Xie & Huang, 2021; Yan & Guo, 2021; Lyu et al., 2021; Li et al., 2021). Recently, based on PML classification, Cole et al. (2021) take a step further and present the paradigm called *Single Positive Labels* (SPL), where the data only provides one correct (positive) label for each instance. A simple example is provided to illustrate the the difference between the PML and SPL paradigms: for an image containing a sofa, a chair, and a potted plant but without a person or a car, for PML paradigm, we are given the label: sofa (yes), chair (yes), *person (no)*, others (unknown), while for SPL paradigm, we only know the label: sofa (yes), others (unknown). In fact, many real-world multi-class datasets are potentially multi-labeled, where SPL algorithms can be used directly to explore their underlying multi-labels. For example, the empirical study by Stock & Cisse (2018) presents the multi-label nature in images of ImageNet dateset (Russakovsky et al., 2015). By using the previous datasets for multi-class problems, SPL can save a massive amount of money, time, and labor resources on data collection.

The algorithmic study of SPL is relatively under-exploited. Cole et al. (2021) first formally define the problem of SPL and proposes the adaptive SPL loss, which shows satisfactory experimental performance. However, it suffers from the shortcoming that one of its hyper-parameters, the average number of positive labels, is hard to define and may vary according to data selection. Subsequently, Verelst et al. (2022) and Zhou et al. (2022) also aim to solve the SPL problem, and introduce the

use of spatial consistency loss and entropy maximization, respectively. However, there exists a great demand for theoretical properties for SPL problems.

An alternative idea for solving SPL problems is to utilize PML classifiers. The first category explores the relationship among labels and models the label correlations (Chen et al., 2019; Durand et al., 2019; Huynh & Elhamifar, 2020). However, this category requires at least two labels per instance and thus is incompatible with the SPL problem. The second category turns the PML problem into solving an optimization problem (Sun et al., 2010; Bucak et al., 2011; Cabral et al., 2011; Xu et al., 2013). However, most of these algorithms only show promising performances on conventional datasets when there are sufficient positive labels per instance. In other words, directly utilizing these algorithms on SPL may induce severe performance degradation. Besides, one can also solve the SPL problem through positive-unlabeled (PU) frameworks for multi-label learning (Sun et al., 2010; Hsieh et al., 2015; Han et al., 2018; Kanehira & Harada, 2016). Nonetheless, these methods are rarely explored in the SPL setting, and most of the works cannot be directly applied to large-scale multi-labeled image classification.

Under such conditions, aiming at solving the SPL problem, we propose a new loss function called *leveraged asymmetric loss with disambiguation (LASD)*, which explicitly copes with the challenges of extreme label imbalance and label self-disambiguation. To address the effectiveness of our proposed loss function, we for the first time use the concept of risk consistency to show the relationship between loss functions for SPL and that for fully supervised multi-label data. The contribution of this paper can be summarized as follows:

- We propose a novel loss function for the SPL problem, where we introduce a pair of leverage parameters to address the severer negative-positive label imbalance that occurred in SPL than in ordinary multi-label learning. Moreover, we resort to the self-labeling mechanism to disambiguate the unobserved labels and alleviate the bad impact of false negatives.

- We for the first time analyze an SPL loss function from the perspective of risk consistency. Under mild sampling assumptions, we first show the theoretical link between arbitrary SPL losses and losses for ordinary multilabel learning. Then we prove the risk consistency of our LASD loss to the cost-sensitive Hamming loss, which guarantees the effectiveness of LASD in dealing with severe label imbalance. This theoretical result also provides theoretical guidance to the choice of the leverage parameters.

- In experiments, we compare our proposed loss with other state-of-the-art SPL methods on multiple multi-label image classification datasets and show the effectiveness of our method. Empirical understandings of the leverage parameter and ablation studies are also conducted.

## 2 RELATED WORKS

### 2.1 METHODS TARGETING AT SPL PROBLEM

Cole et al. (2021) first proposed the idea of SPL, which regards the false-negative labels as label noise, decreases the weights of the negative labels in the loss function, and uses the label smoothing (Szegedy et al., 2016) to trim the parameters. However, their loss function contains a hyper-parameter describing the average number of positive labels for each instance, which is hard to obtain and can vary across datasets. Verelst et al. (2022) utilized a consistency loss to keep feature map output the same across training epochs, making the model for multi-label learning act in synergy with the ubiquitous random resize crop data-augmentation. However, they focus less on the high negative-positive label imbalance in the SPL problem. Moreover, both the works mentioned above failed to justify their proposed methods theoretically. Zhou et al. (2022) for the first time introduced the idea of entropy maximization on the unlabeled data and utilized asymmetric pseudo-labeling to address the negative-positive imbalance. Kim et al. (2022) discussed the loss correction method to solve the SPL problem, while SPL algorithms with sound theoretical understandings from the statistical view are still yet to be exploited.

### 2.2 PML AND PU METHODS FOR SPL PROBLEM

The majority can be summarized in two branches: The first branch establishes a new setting with positive and negative labels and solves the problems by designing efficient loss and optimization

procedures. The other branch pays more attention to label completion, which initially aims at completing unknown labels from the test set. Missing labels on the training set can also be jointly reconstructed in the optimization frameworks. We omit here the discussions of branches incompatible with SPL, including the requirement of at least two positive labels or one positive and one negative label for each instance and the need for a fraction of fully-labeled instances.

**Positive- and negative-label methods.** Sun et al. (2010) formed a convex optimization problem to predict the unknown labels, which assumes the positive instances will be as sparse as possible for each label. Bucak et al. (2011) similarly presented a ranking-based optimization problem using group lasso to obtain the sparse solution of instances for each label. Zhang et al. (2021) redesigned the loss for negative labels to focus on semi-hard samples and introduced self-paced loss correction based on the maximum likelihood criterion.

**Label matrix completion method.** As for label completion methods, Cabral et al. (2011); Xu et al. (2013) constructed a data matrix composed of training and testing features with training labels and then use ranking minimization to complete unknown testing labels. Hsieh et al. (2015) conducted matrix completion in the aspect of PU learning, and Han et al. (2018) used logistic matrix factorization for label matrix reconstruction.

To the best of our knowledge, few theoretical works have been done on the SPL problem, especially from the statistical view. In this work, we propose a new loss function for the SPL problem and for the first time conduct an analysis on the risk consistency of the SPL problem. To be specific, under the mild sampling assumption of labels, we provide the theoretical link between SPL losses and the losses for fully supervised multi-label learning. Based on this result, we are able to provide further theoretical guidance on the choice of the leverage parameter in our proposed loss function.

## 3 PRELIMINARY

Firstly we will formulate the fully observed multi-label setting from the perspective of the positive label set and then introduce the corresponding single-positive-label setting.

### 3.1 FULLY OBSERVED LABELS

In the standard multi-*class* classification problem, each input variable $X$ from the input space $\mathcal{X}$ is assigned a single label from $[K] := \{1, \ldots, K\}$, where $K$ is the number of classes. In the multi-*label* classification problem under the fully observed label setting, each $X$ is associated with a set of positive labels $\vec{Y} \subset [K]$ named the positive label set. For any $y \in [K]$, if the $y$-th class is relevant to $X$, then $y \in \vec{Y}$ and if the $y$-th class is not relevant, then $y \notin \vec{Y}$. Moreover, the positive multi-label space is denoted as $\overline{\mathcal{Y}} := \{\vec{Y} : \vec{Y} \subset [K]\}$. We denote $\overline{P}$ as the probability distribution of the fully labeled data $(X, \vec{Y})$ defined on $\mathcal{X} \times \mathcal{Y}$ and its corresponding probability density function as $\overline{p}$. The goal is to find a function $f : \mathcal{X} \to [0, 1]^K$ that predicts the applicable labels for each $x \in \mathcal{X}$. The formal objective is to find an $f$ that minimizes the risk

$$\bar{\mathcal{R}}(f) := \mathbb{E}_{(X,\vec{Y})\sim\overline{P}} \overline{\mathcal{L}}(f(X), \vec{Y}), \tag{1}$$

where $\overline{\mathcal{L}} : [0, 1]^K \times \overline{\mathcal{Y}} \to \mathbb{R}^+$ is some multi-label metric such as the average precision or 0-1 error.

### 3.2 SINGLE POSITIVE LABEL

In the multi-class classification problem under single positive label setting, each $X$ is associated with only one single positive label $Z \in \mathcal{Z} := [K]$ and obviously $Z$ is one element in the positive label set $\vec{Y}$, i.e. $Z \in \vec{Y}$. We denote $P$ as the probability distribution of the single positive labeled data $(X, Z)$ defined on $\mathcal{X} \times Z$ and its corresponding probability density function as $p(x, z)$. Denote the single positive label loss function as $\mathcal{L} : [0, 1]^K \times [K] \to \mathbb{R}^+$ and its corresponding risk as

$$\mathcal{R}(f) := \mathbb{E}_{(X,Z)\sim P} \mathcal{L}(f(X), Z). \tag{2}$$

### 3.3 RISK CONSISTENCY

In the single positive label setting, we say a single positive label loss function $L$ satisfies the *risk consistency*, if for any function $f$, the fully observed label risk $\bar{\mathcal{R}}(f)$ w.r.t. $\bar{P}$ equals the single positive label risk $\mathcal{R}(f)$ w.r.t $P$, i.e. $\bar{\mathcal{R}}(f) = \mathcal{R}(f)$.

Through the concept of risk consistency, we are able to relate the SPL problem to the standard multi-label classification. To be specific, risk consistency guarantees the empirical risk w.r.t. an SPL loss to be an unbiased estimator of the risk w.r.t. a multi-label loss. Further, if an SPL loss is risk consistent to a multi-label loss, which is proved to be Bayes consistent (Gao & Zhou, 2011), then the SPL loss is also Bayes consistent. In this case, the minimizer of the SPL risk is theoretically guaranteed to be consistent with the Bayes optimum.

## 4 THE PROPOSED LEVERAGED ASYMMETRIC LOSS WITH DISAMBIGUATION

### 4.1 LEVERAGED ASYMMETRIC LOSS

Recall that under the fully supervised multi-label learning, Ridnik et al. (2021) proposed the asymmetric loss (ASL) to address the high negative-positive imbalance problem concomitant with multi-label learning. To be specific, the ASL loss without probability shift is

$$\mathcal{L}_{\text{ASL}}(f(x), \vec{y}) = \sum_{k=1}^{K} \left[ \mathbf{1}_{[k \in \vec{y}]} L_{\gamma_+}(f_k(x)) + \mathbf{1}_{[k \notin \vec{y}]} L_{\gamma_-}(f_k(x)) \right], \tag{3}$$

where we take the binary focal loss

$$\begin{aligned} L_{\gamma_+}(f_k(x)) &= -(1 - f_k(x))^{\gamma_+} \log(f_k(x)), \\ L_{\gamma_-}(f_k(x)) &= -f_k(x)^{\gamma_-} \log(1 - f_k(x)), \end{aligned} \tag{4}$$

and $\gamma_+, \gamma_- \geq 0$.

In order to apply the loss for multi-label learning to the SPL problem, the vanilla idea is to assume that unobserved labels are negative. However, note that the imbalance of fully supervised multi-label learning originates from the existence of more negative labels over positives. In the SPL problem, the negative-positive label is even more imbalanced, since we have only one positive label, and all unobserved labels are assumed to be negative. Therefore, we have to introduce an additional leverage parameter $\lambda$ to tackle the extreme label imbalance, and propose the leveraged asymmetric loss

$$\mathcal{L}_{\text{LAS}}(f(x), z) = \sum_{k=1}^{K} \left[ \mathbf{1}_{[z=k]} L_{\gamma_+}(f_k(x)) + \lambda \cdot \mathbf{1}_{[z \neq k]} L_{\gamma_-}(f_k(x)) \right], \tag{5}$$

where $\lambda \in [0, 1]$.

### 4.2 PSEUDO LABEL DISAMBIGUATION

One thing we should pay attention to is that the assumed negative labels may indeed be positive. Therefore, we attach pseudo labels to the unobserved labels and treat the pseudo positives and pseudo negatives differently. To this end, we reformulate equation 5 with disambiguation, and achieve the *leveraged asymmetric loss with disambiguation* (LASD), i.e.

$$\mathcal{L}_{\text{LASD}}(f(x), z) = \sum_{k=1}^{K} \left[ \mathbf{1}_{[z=k]} \cdot L_{\gamma_+}(f_k(x)) + \lambda_+ \cdot \mathbf{1}_{[z \neq k]} \mathbf{1}_{[f_k(x) > \tau]} \cdot L_{\gamma_+}(f_k(x)) \right.$$

$$\left. + \lambda_- \cdot \mathbf{1}_{[z \neq k]} \mathbf{1}_{[f_k(x) \leq \tau]} \cdot L_{\gamma_-}(f_k(x)) \right], \tag{6}$$

where $\tau \in [0, 1]$ is the disambiguation threshold, $\gamma_+, \gamma_-, \lambda_+, \lambda_- \geq 0$, and $L_{\gamma_+}(\cdot)$ and $L_{\gamma_-}(\cdot)$ are defined in equation 4. In experiments, we select the leverage parameters $\lambda_+$ and $\lambda_-$ based on sound theoretical guarantees. For the sake of readability, we delay the justifications to Theorem 5.4 in Section 5. The LASD loss equation 6 can be decomposed into three terms: the single positive term, the pseudo positive term, and the pseudo negative term.

### 4.3 Consistency Regularization

To further increase the precision of pseudo positive and negative annotations introduced in Section 4.2, we adopt the technique of data augmentation and introduce a consistency regularization to encourage the network's output to be invariant to different randomly augmented images. To be specific, given the original image $x$ and its two random augmentations $x'$ and $x''$, we minimize the $\ell_2$ distance of each output pair. The consistency regularization is therefore formulated as

$$\Phi_{\text{Consist}}(x) = \|f(x) - f(x')\|_2^2 + \|f(x) - f(x'')\|_2^2 + \|f(x') - f(x'')\|_2^2. \tag{7}$$

The final loss function we used is then formulated as

$$\mathcal{L}_{\text{LASD}}(f(x), z) + \alpha * \Phi_{\text{Consist}}(x), \tag{8}$$

where $\alpha$ is the weight parameter used to balance the contributions of the loss term $\mathcal{L}_{\text{LASD}}(f(x), z)$ and the regularization term $\Phi_{\text{Consist}}(x)$.

## 5 Theoretical Analysis

### 5.1 Assumption

The annotator randomly assigns one of the positive labels $\vec{Y}$ to be the *single positive label $Z$*. We formulate this kind of annotation as the following assumption.

**Assumption 5.1** (Sampling of Labels). Let $\bar{p}$ and $p$ be the joint probability density functions of the fully labeled data $(X, \vec{Y})$ and the single positively labeled data $(X, Z)$. Assume that

$$p(x, z) = \sum_{\vec{y} \in \mathcal{Y}_z} \frac{1}{|\vec{y}|} \bar{p}(x, \vec{y}),$$

where $\mathcal{Y}_z = \{\vec{y} : z \in \vec{y}, \vec{y} \in \overline{\mathcal{Y}}\}$.

Note that $P(z|x) = \sum_{\vec{y} \in \mathcal{Y}_z} \bar{P}(\vec{y}|x) P(z|\vec{y}, x)$, where $P$ and $\bar{P}$ denote the corresponding probability distributions of $p$ and $\bar{p}$. Therefore, Assumption 5.1 inherently assumes the single positive label $z$ is uniformly sampled from the fully observed labels, i.e. $P(z|\vec{y}, x) = \frac{1}{|\vec{y}|}$.

Assumption 5.1 is inspired by the conventional modeling of weakly supervised learning problems. For example, in learning from complementary labels, the complementary label is assumed to be independently and uniformly drawn given the true label (Ishida et al., 2017; 2019; Gao & Zhang, 2021); in learning from partial labels, the partial labels are also assumed to be uniformly drawn (Feng et al., 2020). The intuition behind is straightforward: with no prior knowledge given, the simplest and most intuitive way is to assume that the single positive labels are uniformly sampled. Therefore, in this paper, we naturally assume that the single positive label is uniformly sampled, given the ground truth multi-labels.

Here we also propose a practical and efficient way of annotating multi-labeled instances with single positive labels, which could reduce the cost and labor of annotating large datasets, while satisfies the sampling procedure in Assumption 5.1. In order to get samples with the fully observed labels, annotators are often asked to label whether each class of the sample is positive or negative. However, this process is highly time-consuming when the number of classes $K$ is large. Instead, we may choose one of the classes uniformly at random and ask the annotator whether the class is positive or not. Such a yes/no question can be much easier and quicker to be answered than labeling all classes. If the answer is yes, this class will be treated as a single positive label; otherwise, we will select the other class again until a true label is obtained.

### 5.2 Risk Consistency

The concept of risk consistency is a widely used tool in analyzing loss functions for weakly supervised learning problems (see Ishida et al. (2017; 2019); Gao & Zhang (2021)). In learning from single positive labels, we say an SPL loss function $\mathcal{L}$ is risk-consistent to a multi-class loss function $\bar{\mathcal{L}}$ if they have the same classification risk, given the same classifier $f$. Note that risk consistency implies the coincidence of the optimal classifiers, i.e., learning from single positive labels with $\mathcal{L}$ results in the same optimal classifier learned from fully labeled data with $\bar{\mathcal{L}}$.

**Theorem 5.2** (Risk Consistency). *Under Assumption 5.1, if $\mathcal{L}$ and $\bar{\mathcal{L}}$ satisfy*

$$\bar{\mathcal{L}}(f(x), \vec{y}) = \frac{1}{|\vec{y}|} \sum_{z \in \vec{y}} \mathcal{L}(f(x), z),$$ (9)

*then we have risk consistency, i.e.*

$$\bar{\mathcal{R}}(f) = \mathcal{R}(f).$$ (10)

In Theorem 5.2, we derive the general condition of risk consistency under the setting of multi-label learning with single positive labels, which holds for arbitrary loss functions for the SPL problem. To the best of our knowledge, this is the first time that a single positive loss is analyzed from the perspective of risk consistency. By Theorem 5.2, risk consistency requires the average of the single positive loss $\mathcal{L}(f(x), z)$ over all potential positive labels $z \in \vec{y}$ to coincide with the fully supervised multi-label loss $\bar{\mathcal{L}}(f(x), \vec{y})$.

Now we consider the risk consistency of the single positive loss function introduced in Equation equation 6.

**Theorem 5.3.** *If we assume that $|\mathbf{1}_{[f_k(x) > \tau]} - \mathbf{1}_{[k \in \vec{y}]}| \leq \varepsilon$, for some $\varepsilon > 0$, then as $\varepsilon \to 0$, the single positive label loss $\mathcal{L}$ defined in equation 6 is risk consistent to the following fully observed label loss $\bar{\mathcal{L}}$ given by*

$$\bar{\mathcal{L}}(f(x), \vec{y}) = \left( \frac{1}{|\vec{y}|} + \lambda_+ \left(1 - \frac{1}{|\vec{y}|}\right) \right) \sum_{k \in \vec{y}} L_{\gamma_+}(f_k(x)) + \lambda_- \sum_{k \notin \vec{y}} L_{\gamma_-}(f_k(x)).$$ (11)

Theorem 5.3 shows that as the model is nearly well trained and the pseudo labels are nearly correctly labeled, i.e. $\varepsilon \to 0$, the SPL loss defined in equation 6 is risk consistent to the multi-class loss $\bar{\mathcal{L}}$ defined in equation 11, consisting of binary losses on the observed single positive label, the pseudo positive labels, and the pseudo negative labels, respectively.

We show in Theorem 5.3 that our LASD loss is risk consistent to the weighted binary focal loss, which has a similar form to the widely used binary cross-entropy (BCE) loss for multi-label classification. However, equation 11 uses focal loss instead of cross-entropy loss to address the high label imbalance in multi-label learning. Further, when $|\vec{y}| = 1$, where the SPL problem reduced to the ordinary multi-class classification, equation 11 reduces to the one-verses-all loss function (Zhang, 2004).

Next, we prove that our SPL loss defined in equation 6 is risk consistent to the cost-sensitive Hamming loss, and thus can solve the high positive-negative imbalance.

**Theorem 5.4.** *Taking the same assumptions in Theorem 5.3. If $1 + \lambda_+(|\vec{y}| - 1) = \lambda_-(K - |\vec{y}|)$, then as $\varepsilon \to 0$ the single positive label loss $\mathcal{L}$ defined in equation 6 is consistent to the cost-sensitive Hamming loss*

$$\bar{\mathcal{L}}_{\text{ham}} = \sum_{k \in \vec{y}} \frac{1}{|\vec{y}|} \mathbf{1}_{[\hat{y}_k = 1]} + \sum_{k \notin \vec{y}} \frac{1}{K - |\vec{y}|} \mathbf{1}_{[\hat{y}_k = -1]}.$$ (12)

As is known, cost-sensitive learning (Elkan, 2001; Zhou & Liu, 2010) values false-positive classification errors differently from false negatives and is widely applied in solving class-imbalanced classification problems (Japkowicz & Stephen, 2002). In Theorem 5.4, we show that equation 12 is the cost-sensitive version of Hamming loss (Gao & Zhou, 2011; Wu & Zhou, 2017), a widely used loss function for multi-label classification, where the weights for the false positives and false negatives are inversely proportional to the number of positive and negative labels, respectively. In other words, Theorem 5.4 guarantees the effectiveness of our LASD for the SPL problem in solving the severe positive-negative label imbalance by properly selecting the leverage parameters $\lambda_+, \lambda_-$.

## 5.3 DISCUSSIONS

In this part, we compare our LASD with related works for the SPL problem.

Firstly, we show that our proposed LASD loss is an inclusive form, and thus the theoretical analysis in Theorems 5.3 and 5.4 also apply to the previous related works. To be specific, as a special case,

Table 1: Details of four image classification multi-label datasets.

| Dataset | $n_{train}$ | $n_{test}$ | $n_{class}$ | Avg. pos. per Image |
|---|---|---|---|---|
| PASCAL VOC 2012 (VOC12) (Everingham et al.) | 5,717 | 5,823 | 20 | 1.46 |
| MS-COCO 2014 (COCO) (Lin et al., 2014) | 82,081 | 40,137 | 80 | 2.94 |
| NUS-WIDE (NUS) (Chua et al., 2009) | 150,000 | 60,260 | 81 | 1.89 |
| CUB-200-2011 Dataset (CUB) (Wah et al., 2011) | 5,994 | 5,794 | 312 | 31.4 |

when taking $\lambda_+ = \lambda_- = 1$, our LASD loss reduces to the SPLC loss (without probability shifting) proposed by (Zhang et al., 2021). Besides, when further taking $\lambda_+ = \lambda_- = \tau = 1$, our LASD loss reduces to the ASL loss without probability shifting (Ridnik et al., 2021). In order to address the additional high label imbalance brought by the missing labels, SPLC uses the probability shift technique to increase the weight of the semi-hard pseudo positive labels, whereas ASL adopts the probability shift technique to erase the weight of (assumed) easy negatives. By contrast, we propose to use a pair of leverage parameters $\lambda_+$ and $\lambda_-$ to address this problem, which uniformly down weight all unobserved labels (including both pseudo positives and pseudo negatives). In Theorem 5.4, we theoretically prove that properly selected leverage parameters help to balance the weights among the observed single positive label, pseudo positives, and pseudo negatives. Moreover, when taking $\tau = 1$ and $\gamma_+ = \gamma_- = 0$, our LASD loss reduces to the weak assume negative (WAN) loss, which intuitively justifies lowering the weight of the assumed negatives in the binary cross-entropy loss. Nonetheless, as a byproduct of this work, our theory also complements the choice of the parameter in the WAN loss from the perspective of risk consistency.

Secondly, we compare our LASD with EM loss proposed in (Zhou et al., 2022), which also uses pseudo labels to solve the SPL problem. Zhou et al. (2022) only annotate pseudo negatives to avoid the use of noisy negative pseudo-labels and resort to entropy maximization for the rest of the unobserved labels. On the contrary, we use both pseudo positives and pseudo negatives in this work. In Appendix A.3, we empirically verify that when the model is well trained, the precision of pseudo annotations is sufficiently high.

## 6 EXPERIMENTS

### 6.1 DATASET DESCRIPTIONS

We evaluate the performance of our proposed LASD loss on four benchmark multi-label image datasets. We summarize their details in Table 1, including the number of samples for training and for evaluation, and the number of classes. More detailed descriptions are shown in Appendix A.2.1. We generate the single positive multi-label datasets for training by randomly choosing one positive label and discarding other annotations for each training sample from fully labeled multi-label image datasets. The generation process is applied only once per dataset, and the generated single positive multi-label dataset is used for all comparisons on that dataset. We split 20% of the training set for validation. Note that different from the training set, the validation and test sets are fully labeled.

### 6.2 COMPARED METHODS

To evaluate our method on large-scale multi-label image databases, we compare with deep neural network based SPL algorithms and run experiments on their publicly available codes. Specifically, we compare the following loss functions with the formulations of loss functions listed in Appendix A.2.2. $\mathcal{L}_{AN}$ and $\mathcal{L}_{FAN}$ are the vanilla ideas that assume all unobserved labels as negatives in the binary cross-entropy and binary focal loss functions, respectively. $\mathcal{L}_{AN-LS}$ is a variant of $\mathcal{L}_{AN}$ where the label smoothing technique (Szegedy et al., 2016; Müller et al., 2019) is incorporated to tackle the issue of "false-negative", while $\mathcal{L}_{WAN}$ down weights all assumed negatives to tackle this problem. $\mathcal{L}_{ASL}$ is an ordinary multi-label classification method proposed in (Ridnik et al., 2021), where we also assume the unobserved labels to be negative. $\mathcal{L}_{ASL-PS}$ is a variant of $\mathcal{L}_{ASL}$ where the technique of probability shifting is used. $\mathcal{L}_{EPR}$ and $\mathcal{L}_{ROLE}$ (Cole et al., 2021) introduce the expected number of positive labels per image named $k$ as regularization. Different from $\mathcal{L}_{EPR}$, $\mathcal{L}_{ROLE}$ jointly trains

Table 2: The test mAP (%) evaluated on four image classification multi-label datasets. The best results are marked in **bold** and the second best marked in underline.

| Settings | Losses | VOC12 | COCO | NUSWIDE | CUB |
|---|---|---|---|---|---|
| | $\mathcal{L}_{\mathrm{AN}}$ | $85.89 \pm 0.38$ | $64.92 \pm 0.19$ | $42.27 \pm 0.56$ | $18.31 \pm 0.47$ |
| | $\mathcal{L}_{\mathrm{FAN}}$ | $87.03 \pm 0.43$ | $68.24 \pm 0.04$ | $46.22 \pm 0.33$ | $19.42 \pm 0.38$ |
| | $\mathcal{L}_{\mathrm{WAN}}$ | $86.98 \pm 0.36$ | $67.59 \pm 0.11$ | $45.71 \pm 0.23$ | $19.15 \pm 0.56$ |
| | $\mathcal{L}_{\mathrm{AN-LS}}$ | $87.90 \pm 0.21$ | $67.15 \pm 0.13$ | $43.77 \pm 0.29$ | $16.26 \pm 0.45$ |
| Single | $\mathcal{L}_{\mathrm{ASL}}$ | $86.99 \pm 0.05$ | $67.90 \pm 0.18$ | $46.68 \pm 0.22$ | $19.72 \pm 0.12$ |
| Positive | $\mathcal{L}_{\mathrm{ASL-PS}}$ | $87.32 \pm 0.06$ | $68.46 \pm 0.23$ | $46.96 \pm 0.08$ | $20.11 \pm 0.15$ |
| Labeled | $\mathcal{L}_{\mathrm{EPR}}$ | $85.15 \pm 0.58$ | $63.94 \pm 0.12$ | $45.41 \pm 0.05$ | $19.53 \pm 0.19$ |
| | $\mathcal{L}_{\mathrm{ROLE}}$ | $87.77 \pm 0.22$ | $67.04 \pm 0.19$ | $41.63 \pm 0.35$ | $13.66 \pm 0.24$ |
| | $\mathcal{L}_{\mathrm{LL-R}}$ | $89.23 \pm 0.13$ | $71.02 \pm 0.09$ | $47.44 \pm 0.28$ | $19.52 \pm 0.35$ |
| | $\mathcal{L}_{\mathrm{LL-Ct}}$ | $89.06 \pm 0.17$ | $70.51 \pm 0.14$ | $48.04 \pm 0.22$ | $20.43 \pm 0.38$ |
| | $\mathcal{L}_{\mathrm{LL-Cp}}$ | $88.40 \pm 0.24$ | $70.74 \pm 0.08$ | $48.30 \pm 0.11$ | $20.14 \pm 0.37$ |
| | $\mathcal{L}_{\mathrm{EM}}$ | $89.09 \pm 0.17$ | $70.70 \pm 0.31$ | $47.15 \pm 0.11$ | $20.85 \pm 0.42$ |
| | $\mathcal{L}_{\mathrm{EMAPL}}$ | $89.19 \pm 0.31$ | $70.87 \pm 0.23$ | $47.59 \pm 0.22$ | $21.84 \pm 0.34$ |
| | $\mathcal{L}_{\mathrm{LASD}}$ | $\mathbf{89.53 \pm 0.21}$ | $\mathbf{72.22 \pm 0.09}$ | $\mathbf{50.52 \pm 0.03}$ | $\mathbf{22.01 \pm 0.48}$ |

two networks and uses the co-training technique, where one network provides pseudo labels for the supervision of the other network. $\mathcal{L}_{\mathrm{LL-R}}, \mathcal{L}_{\mathrm{LL-Ct}}, \mathcal{L}_{\mathrm{LL-Ct}}$ are three variants of the large loss proposed in Kim et al. (2022), and $\mathcal{L}_{\mathrm{EM}}, \mathcal{L}_{\mathrm{EMAPL}}$ are two variants of the entropy maximization loss proposed in Zhou et al. (2022).

## 6.3 PERFORMANCE EVALUATION

The backbone network is the ImageNet pre-trained ResNet-50 provided in `PyTorch` (Paszke et al., 2019). We add a linear layer after the backbone network. We run experiments on NVIDIA Tesla v100 32GB GPU. We present results under the scenario of fine-tuning the entire network end-to-end for ten epochs. For each method, we search the learning rate in $\{1e-2, 1e-3, 1e-4, 1e-5\}$ and the batch size in $\{8, 16\}$. We select the hyper-parameter whose mean average precision (MAP) is the highest on the validation set. The selection of hyper-parameters for all methods is shown in Appendix A.2.2. We report the test mean average precision (MAP) performance on the multi-label image classification datasets. We repeat the experiments ten times and also report the standard deviations. The best results are marked in **bold** and the second-best are marked in underline.

Table 2 compares the mAPs of our proposed LASD loss with other losses under the SPL paradigm. We conduct significant tests and empirically verify that our proposed LASD outperforms other published methods significantly at the significance level of 0.05 on all four benchmark datasets. Compared with the best performance of other state-of-the-art algorithms for the SPL problem, the improvements of LASD are significant especially on two large multi-label datasets `COCO` and `NUSWIDE`: the LASD loss achieves +1.2 mAP improvement on the `COCO` dataset and +2.2 mAP improvement on the `NUSWIDE` dataset. Compared with gains obtained by former SOTA methods $\mathcal{L}_{\mathrm{LL}}$ (Kim et al., 2022) and $\mathcal{L}_{\mathrm{EMAPL}}$ (Zhou et al., 2022) in their original papers where average +1.4 mAP gains are achieved, the improvements of our LASD method are also remarkable enough. These achievements attribute to the learning strategy of leveraged asymmetric loss with pseudo label disambiguation and consistency regularization.

## 6.4 PARAMETER ANALYSIS

In this part, we conduct parameter analysis on essential parameters of our LASD loss, including the leverage parameter $\lambda_+$, the disambiguation threshold $\tau$, and the strength of the regularization term $\alpha$. We change one parameter while fixing other parameters to be the best parameters chosen from the validation set. Note that the other leverage parameter $\lambda_-$ is theoretically calculated based on Theorem 5.4.

**Parameter Analysis on the leverage parameter $\lambda_+$:** From Figure 1(a) we observe that the performance reaches the best mAP when $\lambda_+ = 0.10$. When $\lambda_+$ becomes too small, the pseudo positive term will be neglected, whereas a large $\lambda_+$ means that we might be overconfident on the pseudo

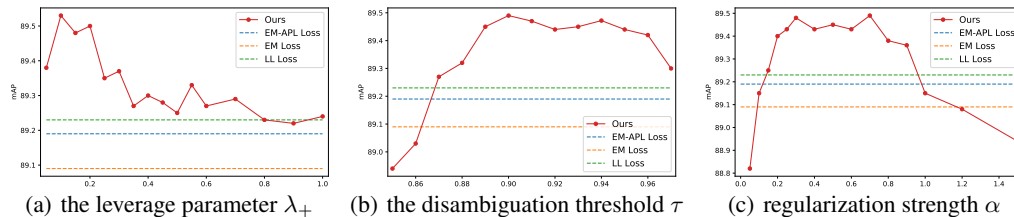

(a) the leverage parameter $\lambda_+$    (b) the disambiguation threshold $\tau$    (c) regularization strength $\alpha$

Figure 1: Parameter analysis on three parameters of our proposed LASD loss on the `VOC12` dataset.

Table 3: Ablation study on the LASD loss on the `VOC12` dataset.

|   | Leverage | Disambiguation | Consistency Regularization | VOC12 |
|---|---|---|---|---|
| A | $\times$ | $\times$ | $\times$ | $85.89 \pm 0.38$ |
| B | $\checkmark$ | $\times$ | $\times$ | $86.99 \pm 0.05$ |
| C | $\checkmark$ | $\checkmark$ | $\times$ | $88.70 \pm 0.14$ |
| D | $\checkmark$ | $\checkmark$ | $\checkmark$ | $89.53 \pm 0.25$ |

positive labels. Note that our method outperforms other competing methods in a wide range of $\lambda_+$, which verifies the robustness of this parameter.

**Parameter Analysis on the disambiguation threshold $\tau$:** Figure 1(b) shows that when $\tau$ is close to 1, it tends to ignore the potential positive labels, while small $\tau$ is at the risk of false positive. For this reason, a suitable $\tau$ slightly smaller than one is beneficial in the SPL problem. Figure 1(b) also verifies the robustness of the selection of $\tau$, as our method outperforms in a wide range of $\tau$.

**Parameter Analysis on the weight parameter of the regularization term $\alpha$:** From Figure 1(c) we observe that our method outperforms other competitors in a wide range between 0.2 to 0.8. When the regularization strength $\alpha$ is close to zero, its performance decreases because the model fails to utilize the power of consistency regularization, whereas when $\alpha$ is too large, the regularization term will be dominated in the loss function and is harmful for the training process.

**Ablation Study:** We discuss the effectiveness of three key components of LASD, including the asymmetric leverage, the pseudo label disambiguation and the regularization term. We compare the following cases: case A) without the leveraged term, the disambiguation term, and the regularization term; case B) with the leveraged term but without the disambiguation term and the regularization term; case C) with the leveraged term and the disambiguation term, but without the regularization term; and case D) with the leveraged term, the disambiguation term, and the regularization term. The results in Table 3 show that all these three terms attribute to the performance improvement. For one thing, leveraging negative and positive samples with two leverage parameter $\lambda_+$ and $\lambda_-$ is essential to tackle the problem of highly negative-positive imbalance problem with the theoretically understanding w.r.t the risk consistency to cost-sensitive Hamming loss. For another, the pseudo label disambiguation further helps to alleviate the false negative problem and the regularization term further increases the reliability of pseudo-labeling.

## 7    CONCLUSION

In this paper, we proposed a novel loss function called leveraged asymmetric loss with disambiguation (LASD) to address the problem of multi-label learning from single positive labels (SPL). In this loss, we introduce a pair of leverage parameters weighting the single positive label and the unobserved labels, so as to deal with the high negative-positive imbalance. Moreover, we incorporate pseudo labels to disambiguate potential positive and negative labels. From the theoretical perspective, we introduce the concept of risk consistency, and derive the link between an arbitrary SPL loss to the losses for ordinary multi-label learning. We prove that our proposed LASD loss is consistent with the cost-sensitive Hamming loss, which can guide the empirical choice of our proposed leverage parameter under sound theoretical guarantees. In experiments, we compare our LASD with other state-of-the-art SPL algorithms, and demonstrate the outstanding performance of our method on multiple real-world datasets.

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

# A APPENDIX

## A.1 PROOFS

*Proof of Theorem 5.2.* We compare the risk in the fully-labeled setting and the risk in the single positive setting as follows.

On the one hand, the risk in fully label setting is

$$
\begin{aligned}
\mathbb{E}_{\bar{P}}\bar{\mathcal{L}}(f(X),\vec{y}) &= \mathbb{E}_X\mathbb{E}_{\vec{Y}|X}\bar{\mathcal{L}}(f(X),\vec{y}) \\
&= \mathbb{E}_X\sum_{\vec{y}\in\mathcal{Y}}\bar{p}(\vec{y}|x)\bar{\mathcal{L}}(f(X),\vec{y})
\end{aligned}
\tag{13}
$$

On the other hand, the risk in the single positive setting is

$$
\begin{aligned}
\mathbb{E}_p\mathcal{L}(f(x),z) &= \mathbb{E}_{X,\vec{Y},Z}\mathcal{L}(f(x),z) \\
&= \mathbb{E}_X\mathbb{E}_{\vec{Y}|X}\mathbb{E}_{Z|\vec{Y}}\mathcal{L}(f(x),z) \\
&= \mathbb{E}_X\mathbb{E}_{\vec{Y}|X}\frac{1}{|\vec{Y}|}\sum_{z\in\vec{Y}}\mathcal{L}(f(x),z) \\
&= \mathbb{E}_X\sum_{y\in\mathcal{Y}}\bar{p}(\vec{y}|x)\cdot\frac{1}{|\vec{y}|}\sum_{z\in\vec{y}}\mathcal{L}(f(x),z).
\end{aligned}
\tag{14}
$$

Combining Equation equation 13 and equation 14, we find out that, if

$$
\bar{\mathcal{L}}(f(x),\vec{y}) = \frac{1}{|\vec{y}|}\sum_{z\in\vec{y}}\mathcal{L}(f(x),z)
\tag{15}
$$

is satisfied, then risk consistency holds, which finishes the proof. $\square$

*Proof of Theorem 5.3.* Combining equation 6 and equation 9, we get the risk consistent fully-observed label loss

$$
\begin{aligned}
\overline{\mathcal{L}}(f(x),\vec{y}) &= \frac{1}{|\vec{y}|}\sum_{z\in\vec{y}}\mathcal{L}(f(x),z) \\
&= \frac{1}{|\vec{y}|}\sum_{z\in\vec{y}}\sum_{k=1}^{K}\left[\left(\mathbf{1}_{[z=k]}+\lambda_+\cdot\mathbf{1}_{[z\neq k]}\mathbf{1}_{[f_k(x)>\tau]}\right)L_{\gamma_+}+\lambda_-\cdot\mathbf{1}_{[z\neq k]}\mathbf{1}_{[f_k(x)\leq\tau]}L_{\gamma_-}\right] \\
&= \frac{1}{|\vec{y}|}\sum_{k=1}^{K}\sum_{z\in\vec{y}}\left[\left(\mathbf{1}_{[z=k]}+\lambda_+\cdot\mathbf{1}_{[z\neq k]}\mathbf{1}_{[f_k(x)>\tau]}\right)L_{\gamma_+}+\lambda_-\cdot\mathbf{1}_{[z\neq k]}\mathbf{1}_{[f_k(x)\leq\tau]}L_{\gamma_-}\right] \\
&= \frac{1}{|\vec{y}|}\sum_{z\in\vec{y}}\left[L_{\gamma_+}(f_z)+\sum_{k=1,k\neq z}^{K}\left(\lambda_+\mathbf{1}_{[f_k(x)>\tau]}L_{\gamma_+}(f_k)+\lambda_-\mathbf{1}_{[f_k(x)\leq\tau]}L_{\gamma_-}(f_k)\right)\right] \\
&= \frac{1}{|\vec{y}|}\sum_{z\in\vec{y}}\left[L_{\gamma_+}(f_z)+\sum_{k=1,k\neq z}^{K}L_\tau(f_k)\right],
\end{aligned}
\tag{16}
$$

where we denote $L_\tau := \lambda_+\mathbf{1}_{[f_k(x)>\tau]}L_{\gamma_+}(f_k)+\lambda_-\mathbf{1}_{[f_k(x)\leq\tau]}L_{\gamma_-}(f_k)$ for notational simplicity. Then we have

$$
\overline{\mathcal{L}}(f(x),\vec{y}) = \frac{1}{|\vec{y}|}\sum_{z\in\vec{y}}L_{\gamma_+}(f_z)+\frac{1}{|\vec{y}|}\sum_{z\in\vec{y}}\left(\sum_{k\notin\vec{y}}L_\tau(f_k)+\sum_{k\in\vec{y},k\neq z}L_\tau(f_k)\right)
$$

$$= \frac{1}{|\vec{y}|} \sum_{z \in \vec{y}} L_{\gamma_+}(f_z) + \frac{1}{|\vec{y}|} \sum_{z \in \vec{y}} \sum_{k \notin \vec{y}} L_{\tau}(f_k) + \frac{1}{|\vec{y}|} \sum_{z \in \vec{y}} \sum_{k \in \vec{y}, k \neq z} L_{\tau}(f_k). \tag{17}$$

Note that $\sum_{z \in \vec{y}} = |\vec{y}|$ and $\sum_{z \notin \vec{y}} = K - |\vec{y}|$. Then the second term of equation 17 becomes

$$\frac{1}{|\vec{y}|} \sum_{z \in \vec{y}} \sum_{k \notin \vec{y}} L_{\tau}(f_k) = \frac{1}{|\vec{y}|} |\vec{y}| \sum_{k \notin \vec{y}} L_{\tau}(f_k) = \sum_{k \notin \vec{y}}^{K} L_{\tau}(f_k). \tag{18}$$

The third term of equation 17 becomes

$$\frac{1}{|\vec{y}|} \sum_{z \in \vec{y}} \sum_{k \in \vec{y}, k \neq z}^{K} L_{\tau}(f_k) = \frac{1}{|\vec{y}|} \sum_{k \in \vec{y}}^{K} \sum_{z \in \vec{y}, z \neq k} L_{\tau}(f_k) = \frac{|\vec{y}| - 1}{|\vec{y}|} \sum_{k \in \vec{y}} L_{\tau}(f_k). \tag{19}$$

Combining equation 17, equation 18, and equation 19, we have

$$\overline{\mathcal{L}}(f(x), \vec{y}) = \frac{1}{|\vec{y}|} \sum_{k \in \vec{y}} L_{\gamma_+}(f_k) + \sum_{k \notin \vec{y}}^{K} L_{\tau}(f_k) + \frac{|\vec{y}| - 1}{|\vec{y}|} \sum_{k \in \vec{y}} L_{\tau}(f_k)$$

$$= \frac{1}{|\vec{y}|} \sum_{k \in \vec{y}} L_{\gamma_+}(f_k) + \sum_{k=1}^{K} L_{\tau}(f_k) - \frac{1}{|\vec{y}|} \sum_{k \in \vec{y}} L_{\tau}(f_k)$$

$$= \frac{1}{|\vec{y}|} \sum_{k \in \vec{y}} L_{\gamma_+}(f_k) + \sum_{k=1}^{K} L_{\tau}(f_k) - \frac{1}{|\vec{y}|} \sum_{k \in \vec{y}} L_{\tau}(f_k)$$

$$= \frac{1}{|\vec{y}|} \sum_{k \in \vec{y}} L_{\gamma_+}(f_k) + \Big( \sum_{k=1}^{K} - \frac{1}{|\vec{y}|} \sum_{k \in \vec{y}} \Big) \Big( \lambda_+ \mathbf{1}_{[f_k(x) > \tau]} L_{\gamma_+}(f_k) + \lambda_- \mathbf{1}_{[f_k(x) \leq \tau]} L_{\gamma_-}(f_k) \Big)$$

$$\tag{20}$$

By assumption, if $\mathbf{1}_{[f_k(x) > \tau]} \geq \mathbf{1}_{[k \in \vec{y}]}$, i.e. $\mathbf{1}_{[f_k(x) > \tau]} - \mathbf{1}_{[k \in \vec{y}]} = \varepsilon$, then there holds

$$\mathbf{1}_{[f_k(x) \leq \tau]} - \mathbf{1}_{[k \notin \vec{y}]} = \big( 1 - \mathbf{1}_{[f_k(x) > \tau]} \big) - \big( 1 - \mathbf{1}_{[k \in \vec{y}]} \big) = \mathbf{1}_{[k \in \vec{y}]} - \mathbf{1}_{[f_k(x) > \tau]} = -\varepsilon. \tag{21}$$

Thus, by equation 20, we have

$$\bar{\mathcal{L}}(f(x), \vec{y}) = \frac{1}{|\vec{y}|} \sum_{k \in \vec{y}} L_{\gamma_+}(f_k) + \Big( \sum_{k=1}^{K} - \frac{1}{|\vec{y}|} \sum_{k \in \vec{y}} \Big)$$

$$\cdot \Big( \lambda_+ \mathbf{1}_{[k \in \vec{y}]} L_{\gamma_+}(f_k) + \lambda_- \mathbf{1}_{[k \notin \vec{y}]} L_{\gamma_-}(f_k) + \varepsilon \lambda_+ L_{\gamma_+}(f_k) - \varepsilon \lambda_- L_{\gamma_-}(f_k) \Big)$$

$$= \frac{1}{|\vec{y}|} \sum_{k \in \vec{y}} L_{\gamma_+}(f_k) + \Big( \sum_{k=1}^{K} - \frac{1}{|\vec{y}|} \sum_{k \in \vec{y}} \Big) \Big( \lambda_+ \mathbf{1}_{[k \in \vec{y}]} L_{\gamma_+}(f_k) + \lambda_- \mathbf{1}_{[k \notin \vec{y}]} L_{\gamma_-}(f_k) \Big)$$

$$+ \Big( \sum_{k=1}^{K} - \frac{1}{|\vec{y}|} \sum_{k \in \vec{y}} \Big) \varepsilon \big( \lambda_+ L_{\gamma_+}(f_k) - \lambda_- L_{\gamma_-}(f_k) \big)$$

$$= \Big( \frac{1}{|\vec{y}|} + \lambda_+ \big( 1 - \frac{1}{|\vec{y}|} \big) \Big) \sum_{k \in \vec{y}} L_{\gamma_+}(f_k(x)) + \lambda_- \sum_{k \notin \vec{y}} L_{\gamma_-}(f_k(x))$$

$$+ \varepsilon \Big( \sum_{k=1}^{K} - \frac{1}{|\vec{y}|} \sum_{k \in \vec{y}} \Big) \big( \lambda_+ L_{\gamma_+}(f_k) - \lambda_- L_{\gamma_-}(f_k) \big) \tag{22}$$

By similar induction, equation 22 also holds for the case where $\mathbf{1}_{[f_k(x) > \tau]} \leq \mathbf{1}_{[k \in \vec{y}]}$. Then as $\varepsilon \to 0$, there holds

$$\bar{\mathcal{L}}(f(x), \vec{y}) \to \Big( \frac{1}{|\vec{y}|} + \lambda_+ \big( 1 - \frac{1}{|\vec{y}|} \big) \Big) \sum_{k \in \vec{y}} L_{\gamma_+}(f_k(x)) + \lambda_- \sum_{k \notin \vec{y}} L_{\gamma_-}(f_k(x)). \tag{23}$$

$\square$

*Proof of Theorem 5.4.* We first consider the loss function

$$\tilde{\mathcal{L}} = \left(\frac{1}{|\vec{y}|} + \lambda_+\left(1 - \frac{1}{|\vec{y}|}\right)\right)\sum_{k\in\vec{y}}\mathbf{1}_{[\hat{y}_k\neq y_k]} + \lambda_-\sum_{k\notin\vec{y}}\mathbf{1}_{[\hat{y}_k\neq y_k]}, \tag{24}$$

which replaces $L_{\gamma_+}$ and $L_{\gamma_-}$ with indicator functions.

Denote $p_{\vec{y}} = \mathrm{P}(\vec{Y} = \vec{y}|x)$. We have the inner risk w.r.t. $\tilde{\mathcal{L}}$

$$\begin{aligned}
W(p,f) &= \sum_{\vec{y}\in\mathcal{Y}} p_{\vec{y}}\tilde{\mathcal{L}}(f(x),\vec{y}) \\
&= \sum_{\vec{y}\in\mathcal{Y}} p_{\vec{y}}\cdot\left(\frac{1}{|\vec{y}|} + \lambda_+\left(1 - \frac{1}{|\vec{y}|}\right)\right)\sum_{k\in\vec{y}}\mathbf{1}_{[\hat{y}_k\neq y_k]} + \sum_{\vec{y}\in\mathcal{Y}} p_{\vec{y}}\cdot\lambda_-\sum_{k\notin\vec{y}}\mathbf{1}_{[\hat{y}_k\neq y_k]} \\
&= \sum_{\vec{y}\in\mathcal{Y}}\sum_{k\in\vec{y}} p_{\vec{y}}\cdot\left(\frac{1}{|\vec{y}|} + \lambda_+\left(1 - \frac{1}{|\vec{y}|}\right)\right)\mathbf{1}_{[\hat{y}_k\neq y_k]} + \sum_{\vec{y}\in\mathcal{Y}}\sum_{k\notin\vec{y}} p_{\vec{y}}\cdot\lambda_-\mathbf{1}_{[\hat{y}_k\neq y_k]} \\
&= \sum_{k\in[K]}\sum_{\vec{y}\in\mathcal{Y}_k} p_{\vec{y}}\cdot\left(\frac{1}{|\vec{y}|} + \lambda_+\left(1 - \frac{1}{|\vec{y}|}\right)\right)\mathbf{1}_{[\hat{y}_k\neq y_k]} + \sum_{k\in[K]}\sum_{\vec{y}\in\mathcal{Y}\setminus\mathcal{Y}_k} p_{\vec{y}}\cdot\lambda_-\mathbf{1}_{[\hat{y}_k\neq y_k]} \\
&= \sum_{k\in[K]}\left[\sum_{\vec{y}\in\mathcal{Y}_k} p_{\vec{y}}\cdot\left(\frac{1}{|\vec{y}|} + \lambda_+\left(1 - \frac{1}{|\vec{y}|}\right)\right)\mathbf{1}_{[\hat{y}_k\neq 1]} + \sum_{\vec{y}\in\mathcal{Y}\setminus\mathcal{Y}_k} p_{\vec{y}}\cdot\lambda_-\mathbf{1}_{[\hat{y}_k\neq -1]}\right]. \tag{25}
\end{aligned}$$

For $k\in[K]$, if for some constant $C>0$, $1 + \lambda_+(|\vec{y}| - 1) = \lambda_-(K - |\vec{y}|) = C$, then we have

$$W(p,f) = C\sum_{k\in[K]}\left[\sum_{\vec{y}\in\mathcal{Y}_k} p_{\vec{y}}\cdot\frac{1}{|\vec{y}|}\mathbf{1}_{[\hat{y}_k=1]} + \sum_{\vec{y}\in\mathcal{Y}\setminus\mathcal{Y}_k} p_{\vec{y}}\cdot\frac{1}{K-|\vec{y}|}\mathbf{1}_{[\hat{y}_k=-1]}\right], \tag{26}$$

and the set of corresponding Bayes predictors of $\tilde{L}$

$$A(p) = \left\{f = f(x) : \hat{y} = F(f) \text{ with } \hat{y}_k = sgn\left(\sum_{\vec{y}\in\mathcal{Y}_k}\frac{1}{|\vec{y}|}p_{\vec{y}} - \sum_{\vec{y}\notin\mathcal{Y}_k}\frac{1}{K-|\vec{y}|}p_{\vec{y}}\right)\right\}. \tag{27}$$

Note that $A(p)$ in equation 27 is the same as the set of Bayes predictors of the cost-sensitive Hamming loss equation 12. By similar induction of Theorem 15 in Gao & Zhou (2011), one can derive that LASD equation 6 with surrogate losses $L_{\gamma_+}, L_{\gamma_-}$ is consistent w.r.t. the cost-sensitive Hamming loss equation 12 if $L_{\gamma_+}, L_{\gamma_-}$ are convex and $\frac{\partial L_{\gamma_+}(f_k)}{\partial f_k}|_{f_k=0} = \frac{\partial L_{\gamma_+}(-f_k)}{\partial f_k}|_{f_k=0} < 0$. By the definition of $L_{\gamma_+}$ and $L_{\gamma_-}$, the above conditions are satisfied and we thus finish the proof. $\square$

## A.2 EXPERIMENTAL DETAILS

### A.2.1 DATASET DESCRIPTION

We provide the details of four benchmark multi-label image datasets below:

- The PASCAL VOC 2012 Dataset (VOC12) (Everingham et al.) includes 5,717 training images and 20 classes. We use the official validation set (5,823 images) as the test set. The VOC2012 data includes images obtained from the "Flickr" website under the "Flickr" terms of use. The identity of the images in the database, e.g., source and name of the owner, has been obscured.
- The MS-COCO 2014 Dataset (COCO) (Lin et al., 2014) contains 82,081 training images and 80 classes. We also use the official validation set (40,137 images) as the test set.
- The NUS-WIDE (NUS) Dataset (Chua et al., 2009) consists of 126,034 and 84,226 images from the official training and test sets respectively. The dataset includes 81 classes. Following the standard practice (Gong et al., 2014; Durand et al., 2019), we combine the official training and test sets and randomly split the whole dataset into the training set with 150,000 images and the non-overlapping test set with 60,260 images. This web image dataset is created by Lab for Media Search at the National University of Singapore and should be only used for non-commercial research or educational purposes.

- The CUB-200-2011 Dataset (CUB) (Wah et al., 2011) contains 5,994 training images and 312 classes. The number of images in the test set is 5,794. Images in this dataset are overlapped with images in ImageNet. The use is restricted to non-commercial research and educational purposes.

We carefully check the descriptions of these data sets and how these data sets were collected to make sure that no personally identifiable information or offensive content is contained.

### A.2.2 COMPARED METHODS

- $\mathcal{L}_{\mathrm{AN}}$ is from the vanilla idea that unobserved labels are assumed negative. Thus, the assumed negative strategy with the binary cross-entropy loss is given by

$$\mathcal{L}_{\mathrm{AN}}(f(x), z) = -\frac{1}{K} \sum_{k=1}^{K} \left[ \mathbf{1}_{[z=k]} \log(f_k) + \mathbf{1}_{[z \neq k]} \log(1 - f_k) \right].$$

- $\mathcal{L}_{\mathrm{FAN}}$ is also from the vanilla idea that unobserved labels are assumed negative. Thus, the assumed negative strategy with the focal loss is given by

$$\mathcal{L}_{\mathrm{FAN}}(f(x), z) = -\frac{1}{K} \sum_{k=1}^{K} \left[ \mathbf{1}_{[z=k]} (1 - f_k)^{\gamma} \log(f_k) + \mathbf{1}_{[z \neq k]} f_k^{\gamma} \log(1 - f_k) \right],$$

where $\gamma = 2$ is suggested in (Lin et al., 2017).

- $\mathcal{L}_{\mathrm{AN-LS}}$ is the label smoothing version to tackle the issue of "false negative". Label smoothing is proposed in (Szegedy et al., 2016; Müller et al., 2019). $\mathcal{L}_{\mathrm{AN-LS}}$ combines the assume negative strategy and the label smoothing training technique. $\mathcal{L}_{\mathrm{AN-LS}}$ is given by

$$\mathcal{L}_{\mathrm{AN-LS}}(f(x), z) = -\frac{1}{K} \sum_{k=1}^{K} [[(1 - \varepsilon/2) \cdot \mathbf{1}_{[z=k]} + \varepsilon/2 \cdot \mathbf{1}_{[z \neq k]}] \cdot \log(f_k) +$$
$$[(1 - \varepsilon/2) \cdot \mathbf{1}_{[z \neq k]} + \varepsilon/2 \cdot \mathbf{1}_{[z=k]} \mathbf{1}_{[z \neq k]}^{\varepsilon/2}] \cdot \log(1 - f_k)],$$

where $\varepsilon$ is the hyper-parameter of label smoothing and is set to $0.1$ as in (Cole et al., 2021).

- $\mathcal{L}_{\mathrm{WAN}}$ is proposed based on the idea that the false negative term in the loss function is down-weighted. $\mathcal{L}_{\mathrm{WAN}}$ is given by

$$\mathcal{L}_{\mathrm{WAN}}(f(x), z) = -\frac{1}{K} \sum_{k=1}^{K} \left[ \mathbf{1}_{[z=k]} \log(f_k) + \lambda \cdot \mathbf{1}_{[z \neq k]} \log(1 - f_k) \right],$$

where $\lambda \in [0, 1]$ is the weight hyper-parameter and is set $\lambda$ to $1/(K-1)$ in practice.

- $\mathcal{L}_{\mathrm{ASL}}$ is also from the vanilla idea that unobserved labels are assumed negative. Thus, the assumed negative strategy with the asymmetric focal loss is given by

$$\mathcal{L}_{\mathrm{ASL}}(f(x), z) = -\frac{1}{K} \sum_{k=1}^{K} \left[ \mathbf{1}_{[z=k]} (1 - f_k)^{\gamma_+} \log(f_k) + \mathbf{1}_{[z \neq k]} f_k^{\gamma_-} \log(1 - f_k) \right],$$

where $\gamma_+ = 0$ and $\gamma_- = 4$ are suggested in (Ridnik et al., 2021).

- $\mathcal{L}_{\mathrm{ASL-PS}}$ is also from the vanilla idea that unobserved labels are assumed negative. $\mathcal{L}_{\mathrm{ASL-PS}}$ is a variant of $\mathcal{L}_{\mathrm{ASL}}$ that probability shifting is added as in (Ridnik et al., 2021). Therefore, denote $p_{m,k}$ as $\max(f_k - m, 0)$, $\mathcal{L}_{\mathrm{ASLPS}}$ is given by

$$\mathcal{L}_{\mathrm{ASLPS}}(f(x), z) = -\frac{1}{K} \sum_{k=1}^{K} \left[ \mathbf{1}_{[z=k]} (1 - f_k)^{\gamma_+} \log(f_k) + \mathbf{1}_{[z \neq k]} p_{m,k}^{\gamma_-} \log(1 - p_{m,k}) \right],$$

where $m$ is the probability shifting parameter. We follow the suggestion in (Ridnik et al., 2021) that $\gamma_+ = 0$, $\gamma_- = 4$, and $m = 0.05$.

- $\mathcal{L}_{\text{EPR}}$ and $\mathcal{L}_{\text{ROLE}}$ are proposed in (Cole et al., 2021). Both of them introduced the expected number of positive labels per image named $k$ as regularization. The penalty is applied at the batch level, and the hyper-parameter $k$ is specified based on the fully labeled training set. Different from $\mathcal{L}_{\text{EPR}}$, $\mathcal{L}_{\text{ROLE}}$ jointly trains two networks using the co-training technique, where one network provides pseudo labels for the supervision of the other network. The expected positive regularization loss $\mathcal{L}_{\text{EPR}}$ is formulated as

$$\mathcal{L}_{\text{EPR}}(\mathbf{F_B}, \mathbf{Z_B}) = \frac{1}{|B|} \sum_{n \in B} \mathcal{L}^+_{\text{BCE}}(\mathbf{f}_n, \mathbf{z}_n) + \lambda R_k(\mathbf{F}_B),$$

where

$$\mathcal{L}^+_{\text{BCE}}(\mathbf{f}_n, \mathbf{z}_n) = -\sum_{k=1}^{L} \mathbf{1}_{[z=k]} \log(f_k),$$

$$R_k(\mathbf{F}_B) = \left( \frac{\hat{k}(\mathbf{F}_B) - k}{L} \right)^2,$$

$$\hat{k}(\mathbf{F}_B) = \frac{\sum_{n \in B} \sum_{i=1}^{L} \mathbf{f}_{ni}}{|B|}.$$

and $f$, $g$ are the image classifier and the label estimator respectively. The regularized online label estimation loss $\mathcal{L}_{\text{ROLE}}$ is defined as

$$\mathcal{L}_{\text{ROLE}} = \frac{\mathcal{L}'\left(\mathbf{F}_B \mid \tilde{\mathbf{Y}}_B\right) + \mathcal{L}'\left(\tilde{\mathbf{Y}}_B \mid \mathbf{F}_B\right)}{2},$$

where

$$\mathcal{L}'\left(\mathbf{F}_B \mid \tilde{\mathbf{Y}}_B\right) = \frac{1}{|B|} \sum_{n \in B} \mathcal{L}_{\text{BCE}}(\mathbf{f}_n, \text{sg}(\tilde{\mathbf{y}}_n)) + \mathcal{L}_{\text{EPR}}(\mathbf{F}_B, \mathbf{Z}_B),$$

and sg is the stop-gradient operation.

- $\mathcal{L}_{\text{LASD}}$ is proposed in our paper. We fix $\gamma_+ = 0$, $\gamma_- = 4$, and search the disambiguation threshold $\tau \in (0.85, 1)$, the leverage parameter $\lambda_+ \in (0, 1]$, and the weight parameter of the regularization term $\alpha$. Another leverage parameter $\lambda_-$ is then calculated according to Theorem 5.4. We provide the final selected hyper-parameters of our proposed LASD as in Table 4.

Table 4: Selected hyper-parameters of our method on each dataset.

|  | VOC12 | COCO | NUS | CUB |
|---|---|---|---|---|
| $\lambda_+$ | 0.10 | 0.03 | 0.015 | 0.005 |
| $\tau$ | 0.90 | 0.90 | 0.95 | 0.93 |
| $\alpha$ | 0.70 | 0.10 | 0.15 | 0.03 |

## A.3 PRECISION OF PSEUDO LABELS

In order to verify that the assumption in Theorem 5.3 that $\mathbf{1}_{[f_k(x)>\tau]} = \mathbf{1}_{[k \in \bar{y}]}$ holds when $f_k(x)$ is trained good enough, we present the averaged Positive Predictive Value (PPV) and the averaged Negative Predictive Value (NPV) of the proposed LASD loss per epoch on the test sets of VOC12 and COCO under the best hyper-parameters. In each epoch, we use the disambiguation threshold $\tau$ to classify the instance $x$ for the $k$-th label: we classify the instance $x$ as positive for the $k$-th label if $f_k(x) > \tau$ and otherwise negative. Then, we calculate the True Positive (TP), False Positive (FP), False Negative (FN), and True Negative (TN). Positive Predictive Value (PPV) is calculated by TP/(TP+FP), and Negative Predictive Value (NPV) is calculated by TN/(TN+FN). We obtained the averaged PPV and NPV for all labels per epoch.

As shown in Figure 2, the PPVs and the NPVs of our LASD loss are signifitantly higher than 90% and even 95%, which means that most of the classified positive labels are true positives and most of the classified negative labels are true negatives. In other words, $\mathbf{1}_{[f_k(x)>\tau]} = \mathbf{1}_{[k \in \bar{y}]}$ and $\mathbf{1}_{[f_k(x) \leq \tau]} = \mathbf{1}_{[k \notin \bar{y}]}$ are empirically verified.

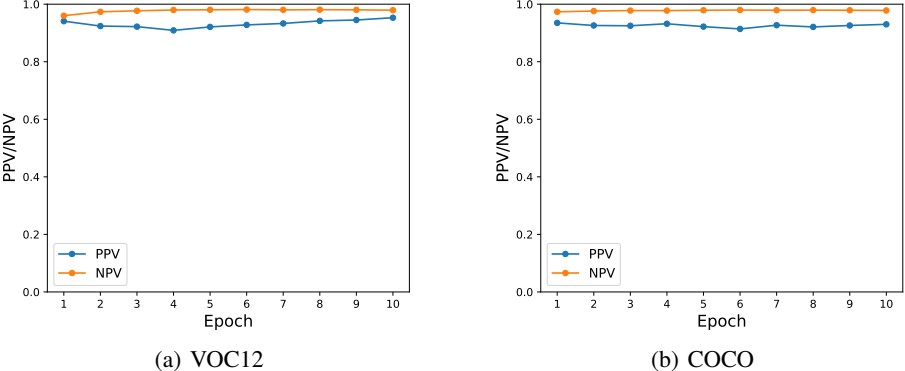

(a) VOC12               (b) COCO

Figure 2: Positive Predictive Value (PPV) and Negative Predictive Value (NPV). The high PPV and NPV results empirically verify the assumption of Theorem 5.3.

