# OpenReview forum: "Leveraged Asymmetric Loss with Disambiguation for Multi-label Recognition with One-Positive Annotations"
_ICLR.cc/2023/Conference — Submitted to ICLR 2023_

### Official Review · Reviewer_HXy7 · 2022-10-19

**Confidence:** 5
**Correctness:** 2
**Technical Novelty And Significance:** 3
**Empirical Novelty And Significance:** 3
**Recommendation:** 3

**Clarity, Quality, Novelty And Reproducibility:**

This paper links the SPL loss with losses for ordinary multi-label classification from the perspective of risk consistency and proves the consistency of the proposed LASD loss to Hamming loss. Extensive experiments are conducted to validate the effectiveness of their method. However, the assumption of Theorem 5.3 that the pseudo-positive labels selected by the threshold are just real positive labels is unreasonable, which makes the proof of the risk consistency unconvincing.

Leveraged asymmetric loss (LAS) has been proposed to alleviate the extreme imbalance of negative-positive labels in the SPL problem. LASD loss just adds a threshold to select the pseudo-positive labels which also has been widely used in previous work, the same is true for consistency regularization. There is not enough innovation in this work.


**Strength And Weaknesses:**

Strengths:
1. A pair of leverage parameters are employed to address the severe negative-positive imbalance of the SPL problem and the usage of consistency regularization can further improve the performance.
2. The proposed method is easy to understand and reproduce.

Weakness:
1. The most significant contribution of this work is analyzing an SPL loss function from the perspective of risk consistency. However, the assumption of Theorem 5.3 is unreasonable, in the inequation $ \left|1_{\left[f_k(x) \geq \tau\right]}-1_{[k \in \vec{y}]}\right| \leq \varepsilon $, the left part only has the value of 0 and 1, letting $\varepsilon \rightarrow 0$ is equivalent to assuming that the pseudo positive labels selected by the threshold are just real positive labels. In this case, the proof of the consistency between LASD loss and Hamming loss makes no sense.
2. Theorem 5.4 guarantees the effectiveness of LASD for the SPL problem in solving the severe positive-negative label imbalance by properly selecting the leverage parameters $\lambda_{+}, \lambda_{-}$. But another import hyper-parameter $\tau$ which is used to select pseudo-positive labels in LASD loss is neglected. It would be better if there is theoretical guidance for the selection of the threshold rather than assuming that the pseudo-positive labels selected by the threshold are just real positive labels.
3. Leveraged asymmetric loss (LAS) has been proposed to alleviate the extreme imbalance of negative-positive labels in the SPL problem. LASD loss just adds a threshold to select the pseudo-positive labels which also has been widely used in previous work. There is not enough innovation in this work.
4. there is a lack of experiments to verify the assumption of Theorem 5.3 and Theorem 5.4.
5. In Table 2, the improvement of the performance on datasets VOC12 and CUB is not obvious enough.


**Summary Of The Paper:**

This paper proposed a novel loss function called leveraged asymmetric loss with disambiguation (LASD) for the problem of multi-label learning from single positive labels (SPL) where only one positive annotation is available. A pair of leverage parameters are employed to address the severe negative-positive imbalance. This paper links the SPL loss with losses for ordinary multi-label classification from the perspective of risk consistency and proves the consistency of the proposed LASD loss to Hamming loss. Finally, experiments including comparing it with other state-of-art methods and ablation are conducted to validate the effectiveness of their method.

**Summary Of The Review:**

A pair of leverage parameters are employed to address the severe negative-positive imbalance of the SPL problem and the usage of consistency regularization can further improve the performance. However, the assumption that the pseudo-positive labels selected by the threshold are just real positive labels is unreasonable and there is a lack of experiments to verify the assumption.

---

> ### Author Response · Authors · 2022-11-20
> **Response to Reviewer HXy7 (2/2)**
>
> **Q4.** There is a lack of experiments to verify the assumption of Theorem 5.3 and Theorem 5.4.
>
> **A4.** *The empirical verification is shown in Figure 2 in Appendix A.3, where the Positive Predictive Value (PPV) and Negative Predictive Value (NPV) are significantly higher than 90% and 95%.*
>
> ---
>
> **Q5.** In Table 2, the improvement of the performance on datasets VOC12 and CUB is not obvious enough.
>
> **A5.** *We conduct significant tests to empirically verify that our proposed LASD achieves highly competitive performance with respect to other published methods. We compare the experimental results of LASD to each of the other methods respectively with two sample t-tests, and calculate the p-values for each pairwise comparison. We find that on all four of the benchmark datasets, our proposed LASD significantly outperforms other listed methods at a significant level of 0.05.*
>
> ---
>
> Thanks for your insightful and constructive comments. Hope our explanations and additional experiments can address your concerns. Please let us know if there is more to clarify. We are happy to take your further questions during the rebuttal stage.

---

> ### Author Response · Authors · 2022-11-20
> **Response to Reviewer HXy7 (1/2)**
>
> We express our sincere gratitude to Reviewer HXy7 for appreciating the methodological presentation of our work. We address your main concerns below.
>
> ---
>
> **Q1.** The most significant contribution of this work is analyzing an SPL loss function from the perspective of risk consistency. However, the assumption of Theorem 5.3 is unreasonable, in the inequation $|1_{[f_k(x)\geq \tau]} - 1_{[k \in \vec{y}]}| \leq \varepsilon$, the left part only has the value of 0 and 1, letting $\varepsilon \to 0$ is equivalent to assuming that the pseudo positive labels selected by the threshold are just real positive labels. In this case, the proof of the consistency between LASD loss and Hamming loss makes no sense.
>
> **A1.** *Thanks for pointing this out, $|1_{[f_k(x)\geq \tau]} - 1_{[k \in \vec{y}]}| \leq \varepsilon$ should be in the sense of expectation, i.e. $\mathbb{E}|1_{[f_k(x)\geq \tau]} - 1_{[k \in \vec{y}]}| \leq \varepsilon$. Moreover, we empirically verify the assumption of Theorem 5.3 in Figure 2 in Appendix A.3, which shows that the Positive Predictive Value (PPV) and Negative Predictive Value (NPV) are significantly higher than 90\% and 95\%.*
>
> ---
>
> **Q2.** Theorem 5.4 guarantees the effectiveness of LASD for the SPL problem in solving the severe positive-negative label imbalance by properly selecting the leverage parameters $\lambda_+,\lambda_-$. But another import hyper-parameter $\tau$ which is used to select pseudo-positive labels in LASD loss is neglected. It would be better if there is theoretical guidance for the selection of the threshold rather than assuming that the pseudo-positive labels selected by the threshold are just real positive labels.
>
> **A2.** *In Theorems 5.3 and 5.4, $\tau$ plays an important role in the assumption that the pseudo labels are correctly annotated. That is, if $\tau$ should be properly chosen such that $\boldsymbol{1}\_{[f_k(x) > \tau]} = \boldsymbol{1}\_{[k \in \vec{y}]}$, then we have our LASD loss risk consistent to the loss $\bar{\mathcal{L}}$ shown in Theorem 5.3. Moreover, the proof of Theorem 5.3 indicates that an error of $\varepsilon$ in the pseudo label annotation results in an error of $\varepsilon \big(\sum_{k=1}^K - \frac{1}{\vert \vec{y} \vert} \sum\_{k\in \vec{y}}\big) \big|\lambda_+ L_{\gamma_+}(f_k) - \lambda_- L_{\gamma_-}(f_k)\big|$ in the risk.*
>
> ---
>
> **Q3.** Leveraged asymmetric loss (LAS) has been proposed to alleviate the extreme imbalance of negative-positive labels in the SPL problem. LASD loss just adds a threshold to select the pseudo-positive labels which also has been widely used in previous work. There is not enough innovation in this work.
>
> **A3.** *Our technical novelty mainly lies in proposing the LASD loss with theoretical guarantees of risk consistency.*
>
> - *Our most significant theoretical contribution is to **analyze SPL loss functions from the perspective of risk consistency**. We introduce the concept of risk consistency, a widely used tool in analyzing loss functions for weakly supervised learning problems, to the SPL problem. To be specific, in Theorem 5.2, we provide the general result on the risk consistency of arbitrary SPL losses. In Theorems 5.3 and 5.4, we apply risk consistency to analyze our LASD loss, and provide the theoretical guidance to the choice of leverage parameters $\lambda_+$ and $\lambda_-$ in our proposed LASD loss.*
> - *The methodology novelty lies in both the introduction of the leverage parameter $\lambda_+$, $\lambda_-$ (in both LAS and LASD) and the pseudo label disambiguation (in LASD). More importantly, **the relationship of $\lambda_+$ and $\lambda_-$ is theoretically guaranteed**. That is, we not only propose the LASD loss function for the SPL problem, but we are also able to tell the optimal choice of parameters in our proposed loss through theoretical results, which is novel in the studies of SPL problems and even in the multi-label learning community.*

---

> > ### Comment · Reviewer_HXy7 · 2022-12-10
> > **Thanks for the response.**
> >
> > Thanks for the response. The main concern about the assumption of Theorem 5.3 has not been solved and the risk consistency is not clearly stated.

---

### Official Review · Reviewer_PKX1 · 2022-10-20

**Confidence:** 5
**Correctness:** 2
**Technical Novelty And Significance:** 2
**Empirical Novelty And Significance:** 2
**Recommendation:** 3

**Clarity, Quality, Novelty And Reproducibility:**

As far as I know, this paper is not the first work to solve SPL problem from the perspective of risk consistency [One Positive Label is Sufficient: Single-Positive Multi-Label Learning with Label Enhancement. Xu, et al.]

**Strength And Weaknesses:**

Strengths:
1. The topic is interesting.
2. The method is easy to understand.
3. Although unsuccessful, the theoretical analysis of the proposed method is to be encouraged.

Weaknesses:
1. The theoretical analysis is deeply flawed. This paper proves the risk consistency of LASD by proving the risk of the classifier learned with LASD approaches to the risk of the classifier **learned with a specifically defined and  (most likely, at least, unproven) non-classification-calibrated multi-label loss function**. Such "consistency" is meaningless because classification-calibrated is the most basic requirement for a loss function.
2. The assumption of Theorem 5.3 is unrealistic. It assumes that if the classifier f can already successfully classify (i.e., the optimal classifier), then LASD is spurious risk-consistent. One wants the learning to be consistent so that the learned classifier is guaranteed to be (asymptotically) equivalent to the optimal classifier, rather than assuming that the classifier is the optimal one first and then proving something else. It's a complete reversal of cause and effect.
3. Assumption 5.1 and Theorem 5.2 are disconnected from the main content. They are neither an original contributions nor relevant to the approach presented in the paper.

**Summary Of The Paper:**

This paper seeks to study a weakly supervised problem called Single Positive Labels (SPL) in multi-label learning where only one positive label is observed for each examples. It proposes a loss function named leveraged asymmetric loss with disambiguation (LASD) following multi-label version of focal loss for SPL problem, and tries to prove that learning with the proposed loss is consistent.

**Summary Of The Review:**

The theoretical analysis of the paper needs further revision and improvement.

---

> ### Author Response · Authors · 2022-11-20
> **Response to Reviewer PKX1**
>
> We express our sincere gratitude to Reviewer PKX1 for appreciating the topic and presentation of our work. We address your main concerns below.
>
> ---
>
> **Q1.** The theoretical analysis is deeply flawed. This paper proves the risk consistency of LASD by proving the risk of the classifier learned with LASD approaches to the risk of the classifier learned with a specifically defined and (most likely, at least, unproven) non-classification-calibrated multi-label loss function. Such "consistency" is meaningless because classification-calibrated is the most basic requirement for a loss function.
>
> **A1.** *It might be important for a loss to be consistent to e.g. the standard Hamming loss when label is **balanced**. However, this is often not true when label is **imbalanced**. For a simple example, in an extremely imbalanced binary classification, one can simply predict all samples as the majority class to achieve a high classification accuracy, whereas this is meaningless since all minority labels are wrongly predicted. A cost-sensitive (weighted) loss function can shift the classification boundary towards the minority class, and therefore result in a more satisfactory classifier. As is shown in Table 1, the percentages of positive labels of the multi-label datasets are about $2\%$ to $10\%$, which is **heavily label imbalanced**. Therefore, it is important to guarantee the consistency of our LASD loss to the cost-sensitive Hamming loss rather than the standard Hamming loss as discussed in Theorem 5.4.*
>
> ---
>
> **Q2.** The assumption of Theorem 5.3 is unrealistic. It assumes that if the classifier f can already successfully classify (i.e., the optimal classifier), then LASD is spurious risk-consistent. One wants the learning to be consistent so that the learned classifier is guaranteed to be (asymptotically) equivalent to the optimal classifier, rather than assuming that the classifier is the optimal one first and then proving something else. It's a complete reversal of cause and effect.
>
> **A2.** *The focus of Theorem 5.3 is different from what is discussed in **Q4**. Theorems 5.3 and 5.4 guarantee the LASD loss (for the SPL problem) reaches the same optimal classifier as the cost-sensitive Hamming loss (for multi-label classification) when the model is well trained. They reveal the equivalence between the optimal classifiers for **SPL classification** and **standard multi-label classification**, instead of studying whether an **SPL loss** is consistent to some **optimal SPL loss**. Moreover, we empirically verify the assumption of Theorem 5.3 in Figure 2 in Appendix A.3, which shows that the Positive Predictive Value (PPV) and Negative Predictive Value (NPV) are significantly higher than 90\% and 95\%.*
>
> ---
>
> **Q3.** Assumption 5.1 and Theorem 5.2 are disconnected from the main content. They are neither an original contributions nor relevant to the approach presented in the paper.
>
> **A3.** *By Assumption 5.1, Theorem 5.2 presents a general result of risk consistency for the SPL problem, which is a widely used tool in analyzing loss functions for weakly supervised learning problems. To be specific, by risk consistency, we are able to show the relationship between an SPL loss and some loss for multi-label learning. It is also the critical result for deriving the following Theorems 5.3 and 5.4.*
>
> ---
>
> **Q4.** As far as I know, this paper is not the first work to solve SPL problem from the perspective of risk consistency [One Positive Label is Sufficient: Single-Positive Multi-Label Learning with Label Enhancement. Xu, et al.]
>
> **A4.** *Xu et. al. is accepted to NeurIPS 2022, whose camera ready version is released on **12 Oct. 2022**, whereas the submission deadline for ICLR 2023 is **28 Sep. 2022**. Therefore we think it is acceptable to claim that this work is the first to solve SPL problem from the perspective of risk consistency in our originally submitted manuscript. Moreover, we use an entirely different approach to analyze the risk consistency of SPL classification from Xu et. al. Therefore, we are able to for the first time prove the **risk consistency for arbitrary SPL losses**, whereas Xu et. al. only provide the risk consistency result for SPL loss with a specific form.*
>
> ---
>
> Thanks for your insightful and constructive comments. Hope our explanations and additional experiments can address your concerns. Please let us know if there is more to clarify. We are happy to take your further questions during the rebuttal stage.

---

> > ### Comment · Reviewer_PKX1 · 2022-12-04
> > **Thanks for the response**
> >
> > Thank you for the response. Unfortunately, the defects I see in the paper remains unresolved.
> >
> > I disagree with the authors' rebuttal to the first two questions. The authors simply do not know what "risk consistency" really means. Weakly-supervised learning methods aim to estimate the classification risk from training data drawn from a distribution different from the true data-generating distribution, and risk-consisitent estimators make it theoretically feasible by rewriting the original $\ell$ to $\bar{\ell}$. Such technique is generally loss-independent, or at best has some mild constraints on $\ell$, but $\ell$ should at least be meaningful in supervised learning. This process is not relevant to whether $p(x,y)$ is "balanced" or not, as it is available for the true risk. Defining a meanless $\ell$ to make $\bar{\ell}$ consistent is absurd, not to mention that it rests on an impossible assumption ("the model is well trained") that has been also proposed by other reviewers.
> > The authors also acknowledge that Assumption 5.1 and Theorem 5.2 are contributions to other works, but this is not explicitly stated in the paper. And I don't think they have anything to do with Theorems 5.3 and 5.4 either.

---

### Official Review · Reviewer_k1k2 · 2022-10-23

**Confidence:** 4
**Correctness:** 3
**Technical Novelty And Significance:** 3
**Empirical Novelty And Significance:** 2
**Recommendation:** 5

**Clarity, Quality, Novelty And Reproducibility:**

As I stated earlier, the presentation is clear and the work appears to be reproducible. Also as I stated earlier, the novelty is incremental. So overall the quality of this work needs further justification.

**Strength And Weaknesses:**

Strengths:
-	the presentation is clear
-	the proposed solution appears to be correct
-	the theoretic analysis appears to make sense
-	the evaluations as well as the ablation studies appear to be extensive
Weaknesses:

I have two major concerns with this work. First, the novelty. The proposed loss function is an extension from the existing LAS loss. While there is arguably something new with the extension, novelty-wise it is incremental at best.

Second, the proposed consistency regularization, while conceptually it makes sense, lacks justification in terms of the current form. Why do we have to consider two augmented samples, not one, or not three? It appears to be rather arbitrary and ad hoc.


**Summary Of The Paper:**

This paper addresses the single positive labels problem in multi-label learning. A new loss function called leveraged asymmetric loss with disambiguation (LASD) is proposed, which is an improved version of the existing leveraged asymmetric loss (ASL) with an explicit pseudo label disambiguation. The proposed algorithm further adds a consistency regularization to the LASD loss. Theoretic analysis provides a link between the proposed LASD and Hamming loss. Empirical evaluations are reported using four benchmark datasets against the state-of-the-art literature.

**Summary Of The Review:**

This work addresses the single positive labels problem in multi-label learning. The paper proposes a new loss function explicitly addressing the asymmetric imbalance between positive and negative labels. The theoretical analysis makes sense and the proposed method is supported by evaluations. I have concern on the novelty and the method.

---

> ### Author Response · Authors · 2022-11-20
> **Response to Reviewer k1k2**
>
> We express our sincere gratitude to Reviewer k1k2 for appreciating the theoretical analysis and extensive evaluation of our work. We address your main concerns below.
>
> ---
>
> **Q1.** The proposed loss function is an extension from the existing LAS loss. While there is arguably something new with the extension, novelty-wise it is incremental at best.
>
> **A1.** ***Both LASD and LAS are novel methods proposed in this paper.** The methodology novelty lies in both the introduction of the leverage parameter $\lambda_+$, $\lambda_-$ (in both LAS and LASD) and the pseudo label disambiguation (in LASD). More importantly, the relationship between parameters $\lambda_+$ and $\lambda_-$ is theoretically guaranteed. That is, we not only propose the LASD loss function for the SPL problem, but we are also able to tell the optimal choice of parameters in our proposed loss through theoretical results, which is novel in the studies of SPL problems and even in the multi-label learning community.*
>
> *Besides, our most significant theoretical contribution is to analyze SPL loss functions from the perspective of risk consistency. We introduce the concept of risk consistency, a widely used tool in analyzing loss functions for weakly supervised learning problems, to the SPL problem. To be specific, in Theorem 5.2, we provide the general result on the risk consistency of arbitrary SPL losses. In Theorems 5.3 and 5.4, we apply risk consistency to analyze our LASD loss, and provide the theoretical guidance to the choice of leverage parameters $\lambda_+$ and $\lambda_-$ in our proposed LASD loss.*
>
> ---
>
> **Q2.** Why do we have to consider two augmented samples, not one, or not three? It appears to be rather arbitrary and ad hoc.
>
> **A2.** *We adopt two augmented samples out of considerations on both effectiveness and efficiency. On the one hand, more augmented samples provide stronger consistency regularization, and thus tend to promote higher accuracy. On the other hand, more augmented samples also require heavier computational burden. When selecting two augmented samples, we are able to achieve 1) consistency between augmented sample and original sample, and 2) consistency between two augmented samples, whereas introduce as less as possible computational burden.*
>
> ---
>
> Thanks for your insightful and constructive comments. Hope our explanations and additional experiments can address your concerns. Please let us know if there is more to clarify. We are happy to take your further questions during the rebuttal stage.

---

### Official Review · Reviewer_k9E5 · 2022-10-26

**Confidence:** 2
**Correctness:** 3
**Technical Novelty And Significance:** 2
**Empirical Novelty And Significance:** 4
**Recommendation:** 6

**Clarity, Quality, Novelty And Reproducibility:**

The paper is clearly written and easy to follow. The quality of the paper is high, especially considering the sufficient experimental results provided. One concern could be the theory part. Specially, Theorem 5.2 is correct, however, it does add value to the study, since this is a too trivial result. Theorem 5.3 and Theorem 5.4, should have added value to the paper, especially considering the equivalence of the proposed loss function to the weighted hamming loss. However, the condition only enforces the positive label (f_k(x)) to be correctly predicted but did not imply anything about the negative labels (f_k(x)<threshold). Without conditions on the correct prediction of negative labels, I cannot be convinced that the proposed loss functions depending on negative labels could have the provided theoretical property. I would like to see more evidence on this part.

Some minor comments
1. f_k(x) may not be defined in the paper before using it.
2. Due to the randomness in generating the data, it is better to generate multiple datasets instead of one to reduce the impact of randomness
3. The paper has provided theoretical results showing the equivalence to weighted hamming loss but the experiments test only MAP. It is better to provide results on more metrics, especially those closely related to the theoretical properties of the proposed method.

**Strength And Weaknesses:**

Strength
1. It proposes an effective loss function for learning from multi-labeled data with only one positive annotation.
2. It has shown sufficient experimental results and overall, the performance of the proposed method is superior
3. The paper is clearly written and easy to follow

Weakness
1. The proposed technique is a combination of existing techniques, may lack technical novelty
2. The theoretical results may not be convincing (will explain in detail later)

**Summary Of The Paper:**

This paper proposes a loss function to learn from multi-label data with only one-positive annotation. Specially, the proposed loss function considers three factors: i) the existing asymmetric loss; ii) it uses a threshold to differentiate true-positive and true-negative in the unlabeled candidate labels. The self-labeled data is further used in the asymmetric loss; iii) a regularizer which ensures the data augmentation will not change the model too much. These items are weighted and added together. The paper analyzes some theoretical properties of the problem and the proposed loss function, and finally, shows sufficient experimental results to demonstrate the effectiveness of the proposed method.

**Summary Of The Review:**

The paper has shown a simple, straightforward, and effective solution to a kind of weakly supervised multi-label learning problem. The studied problem is important and the solution is empirically effective, although the proposed method is not novel. The theoretical results may also need to be strengthened.

---

> ### Author Response · Authors · 2022-11-20
> **Response to Reviewer k9E5 (2/2)**
>
> **Q5.** Due to the randomness in generating the data, it is better to generate multiple datasets instead of one to reduce the impact of randomness
>
> **A5.** *We re-generate the training data and re-conduct the experiments on the VOC dataset. As is shown in the following table, our LASD outperforms the other compared methods.*
>
> | Losses | mAP | Hamming Loss | Weighted Hamming Loss |
> | :---- | :---: | :---: | :---: |
> | $\mathcal{L}_{AN}$ | $85.34 \pm 0.38$ | $(2.653 \pm 0.037) \times 10^{-2}$ | $(6.994 \pm 0.148) \times 10^{-3}$ |
> | $\mathcal{L}_{FAN}$ | $87.12 \pm 0.28$ | $(2.486 \pm 0.054) \times 10^{-2}$ | $(6.551 \pm 0.172) \times 10^{-3}$ |
> | $\mathcal{L}_{WAN}$ | $86.73 \pm 0.52$ | $(2.419 \pm 0.049) \times 10^{-2}$ | $(6.568 \pm 0.157) \times 10^{-3}$ |
> | $\mathcal{L}_{AN-LS}$ | $87.50 \pm 0.36$ | $(2.351 \pm 0.049) \times 10^{-2}$ | $(6.139 \pm 0.141) \times 10^{-3}$ |
> | $\mathcal{L}_{ASL}$ | $87.30 \pm 0.37$ | $(2.443 \pm 0.057) \times 10^{-2}$ | $(6.339 \pm 0.197) \times 10^{-3}$ |
> | $\mathcal{L}_{ASL-PS}$ | $87.36 \pm 0.30$ | $(2.423 \pm 0.028) \times 10^{-2}$ | $(6.334 \pm 0.083) \times 10^{-3}$ |
> | $\mathcal{L}_{EPR}$ | $85.25 \pm 0.61$ | $(2.534 \pm 0.027) \times 10^{-2}$ | $(7.084 \pm 0.091) \times 10^{-3}$ |
> | $\mathcal{L}_{ROLE}$ | $88.24 \pm 0.16$ | $(2.207 \pm 0.047) \times 10^{-2}$ | $(5.826 \pm 0.121) \times 10^{-3}$ |
> | $\mathcal{L}_{LL-R}$ | $87.63 \pm 0.18$ | $(2.281 \pm 0.045) \times 10^{-2}$ | $(5.965 \pm 0.086) \times 10^{-3}$ |
> | $\mathcal{L}_{LL-Ct}$ | $87.37 \pm 0.16$ | $(2.337 \pm 0.028) \times 10^{-2}$ | $(6.169 \pm 0.102) \times 10^{-3}$ |
> | $\mathcal{L}_{LL-Cp}$ | $87.33 \pm 0.53$ | $(2.306 \pm 0.070) \times 10^{-2}$ | $(6.072 \pm 0.170) \times 10^{-3}$ |
> | $\mathcal{L}_{EM}$ | $89.06 \pm 0.29$ | $(2.026 \pm 0.038) \times 10^{-2}$ | $(5.534 \pm 0.097) \times 10^{-3}$ |
> | $\mathcal{L}_{EM-APL}$ | $89.17 \pm 0.22$ | $(2.023 \pm 0.018) \times 10^{-2}$ | $(5.516 \pm 0.053) \times 10^{-3}$ |
> | $\mathcal{L}_{LASD}$ | $\mathbf{89.51 \pm 0.11}$ | $\mathbf{(2.019 \pm 0.032) \times 10^{-2}}$ | $\mathbf{(5.508 \pm 0.077) \times 10^{-3}}$ |
>
> ---
>
> **Q6.** The paper has provided theoretical results showing the equivalence to weighted hamming loss but the experiments test only MAP. It is better to provide results on more metrics, especially those closely related to the theoretical properties of the proposed method.
>
> **A6.** *We run additional experimental comparisons w.r.t. Hamming Loss and Weighted Hamming Loss on the VOC dataset. The results are shown in the table of **A5**. Our LASD outperforms the other methods under both criteria.*
>
> ---
>
> Thanks for your insightful and constructive comments. Hope our explanations and additional experiments can address your concerns. Please let us know if there is more to clarify. We are happy to take your further questions during the rebuttal stage.

---

> ### Author Response · Authors · 2022-11-20
> **Response to Reviewer k9E5 (1/2)**
>
> We express our sincere gratitude to Reviewer k9E5 for appreciating the effectiveness and the superior performance of our work. We address your main concerns below.
>
> ---
>
> **Q1.** The proposed technique is a combination of existing techniques, may lack technical novelty
>
> **A1.** *Our technical novelty mainly lies in proposing the LASD loss with theoretical guarantees of risk consistency.*
>
> - *Our most significant theoretical contribution is to **analyze SPL loss functions from the perspective of risk consistency**. We introduce the concept of risk consistency, a widely used tool in analyzing loss functions for weakly supervised learning problems, to the SPL problem. To be specific, in Theorem 5.2, we provide the general result on the risk consistency of arbitrary SPL losses. In Theorems 5.3 and 5.4, we apply risk consistency to analyze our LASD loss, and provide the theoretical guidance to the choice of leverage parameters $\lambda_+$ and $\lambda_-$ in our proposed LASD loss.*
> - *The methodology novelty lies in both the introduction of the leverage parameter $\lambda_+$, $\lambda_-$ (in both LAS and LASD) and the pseudo label disambiguation (in LASD). More importantly, **the relationship of $\lambda_+$ and $\lambda_-$ is theoretically guaranteed**. That is, we not only propose the LASD loss function for the SPL problem, but we are also able to tell the optimal choice of parameters in our proposed loss through theoretical results, which is novel in the studies of SPL problems and even in the multi-label learning community.*
>
> ---
>
> **Q2.** Theorem 5.2 is correct, however, it does add value to the study, since this is a too trivial result.
>
> **A2.** *Theorem 5.2 presents a general result of risk consistency for the SPL problem, which is a widely used tool in analyzing loss functions for weakly supervised learning problems. To be specific, by risk consistency, we are able to show the relationship between an SPL loss and some loss for multi-label learning. It is also the critical result for deriving the following Theorems 5.3 and 5.4.*
>
> ---
>
> **Q3.**  Theorem 5.3 and Theorem 5.4. However, the condition only enforces the positive label (f_k(x)) to be correctly predicted but did not imply anything about the negative labels (f_k(x) < threshold). Without conditions on the correct prediction of negative labels, I cannot be convinced that the proposed loss functions depending on negative labels could have the provided theoretical property.
>
> **A3.** *The condition corresponding to negative labels can be directly derived from that of the positives. Specifically, $\boldsymbol{1}\_{[f_k(x) \leq \tau]} = 1 - \boldsymbol{1}\_{[f_k(x) > \tau]}$ and $\boldsymbol{1}\_{[k \notin \vec{y}]} = 1 - \boldsymbol{1}\_{[k \in \vec{y}]}$, and therefore we also have $|\boldsymbol{1}\_{[f_k(x) \leq \tau]} - \boldsymbol{1}\_{[k \notin \vec{y}]}| \leq \varepsilon$ by assuming $|\boldsymbol{1}\_{[f_k(x) > \tau]} - \boldsymbol{1}\_{[k \in \vec{y}]}| \leq \varepsilon$.*
>
> ---
>
> **Q4.** f_k(x) may not be defined in the paper before using it.
>
> **A4.** *Recall that $f$ is defined as $f : \mathcal{X} \to [0,1]^K$. $f_k(x)$ is defined as the $k$-th element of $f(x) := \{f_k(x)\}_{k=1}^K$.*

---

### Decision · Program_Chairs · 2023-01-20

**Decision:**

Reject

**Justification For Why Not Higher Score:**

There is only one score above the bar, but even this reviewer has pointed out several issues in the paper. The theoretical part is not in the mature state, contains several issues, and the results might not be of any importance as the authors show that the introduced loss is unbiased for a very specific loss, not really widely used. Interestingly, they claim the opposite, but even the cited papers do not consider this loss function (at least I did not find it in these papers).

**Justification For Why Not Lower Score:**

N/A

**Metareview: Summary, Strengths And Weaknesses:**

The paper considers the problem of multi-label classification from single positive labels. The authors assume that the single label is uniformly drawn from fully observed labels. For this assumption, they study the unbiasedness of introduced loss functions. The experimental studies illustrate the advantages of the proposed approach.

Unfortunately, the theoretical contribution has several drawbacks. The term risk consistency is somehow mixed with the statistical consistency usually considered in the context of learning reductions and surrogate losses. Here, the risk consistency is related to noisy labels and unbiased losses. This is not clearly stated in the paper. The notation and theoretical results need thorough revision as several issues have been pointed out by the reviewers. The current results might be still recovered, but it is not clear whether they are of any real importance. The cost-sensitive Hamming loss is not really widely used. It is not clear why the authors claim the opposite.

The authors might be interested in the analysis of the pick-one label approach given in "Multilabel reductions: what is my loss optimising?" or "A no-regret generalization of hierarchical softmax to extreme multi-label classification".